# A Data-Driven Based Spatiotemporal Model Reduction for Microwave Heating Process with the Mixed Boundary Conditions

**Jiaqi Zhong** [1],* and **Shan Liang** [2]

1   College of Automation, Chongqing University of Posts and Telecommunications, Chongqing 400065, China
2   Key Laboratory of Dependable Service Computing in Cyber Physical Society Ministry of Education, Chongqing University, Chongqing 400044, China; lightsun@cqu.edu.cn
*   Correspondence: plusingzhong@163.com

**Abstract:** In this paper, a data-driven based spatiotemporal model reduction approach is proposed for predicting the temperature distribution and developing the computation speeds in the microwave heating process. Due to the mixed boundary conditions, it is difficult for the traditional spectral method to directly obtain the analytical eigenfunctions. Motivated by the time/space separation theory, we first propose a general framework of spatiotemporal model reduction, which can effectively develop the computation speeds in the numerical analysis of multi-physical fields. Subsequently, the empirical eigenfunctions are generated by applying the Karhunen–Loève theory to decompose the snapshots. Then, the partial differential Equation (PDE) model is discretized into a class of recursive equations and transformed as the reduced-order ordinary differential Equation (ODE) model. Finally, the effectiveness and superiority of the proposed approach is demonstrated by a comparison study with a traditional method on the microwave heating Debye medium.

**Keywords:** spatiotemporal model reduction; microwave heating; mixed boundary conditions; data-driven method





## 1. Introduction

In recent years, microwave heating has been widely applied in the industrial and domestic fields, such as food heating [1], lignite drying [2], effluent oil recovery [3] and materials processing [4]. As a typical volumetric heating method, high frequency microwaves can cause the realignment and internal friction of molecules, which will directly transform the electromagnetic energy into heat energy. Compared with the traditional heating method, microwave energy can improve the heating efficiency, shorten processing time and protect the environment [5,6]. Just as a coin has two sides, it is difficult to avoid hotspots or thermal runaway [7,8] due to the non-uniform electromagnetic distribution in the resonant cavity. Ideally, the microwave heating process should be operated within the expected temperature profiles by optimizing the resonant cavity, mixing with a stirring rod and adjusting the incident power, whose precondition is to deeply understand the fundamental mechanism between the electromagnetic and thermodynamic fields [9]. Therefore, the analysis of the microwave heating model is the popular research interests and challenges in the application of microwave energy field.

Microwave heating process is a typical distributed parameter system, which can vary both temporally and spatially. The temperature rising curves in the different positions may have significant differences [10] due to the energy dissipation and thermal transportation. Generally speaking, Maxwell's equations [11] can be used for describing the propagation of electromagnetic field. Heat transport equations with the boundary and initial conditions can be used for the thermal conductivity and convection. With the help of Poynting's theorem and temperature-dependent permittivity, we can obtain the simultaneous equations,

which consist the partial differential equations (PDEs) and ordinary differential equations (ODEs), to describe the coupling process of multi-physical fields [12]. In order to analyze the thermodynamic behavior and restrain the thermal runaway, many researchers have proposed and improved the different numerical methods, such as, finite element method (FEM) [13] and finite-different time-domain (FDTD) method [14]. These traditional methods usually transform the microwave heating model into the thousands of ODEs [15], which can solve a variety of numerical problems, such as, time-varying parameters, complex boundary conditions and irregular solution domains. However, several global parameters are inherently divided into the finite local parameters by applying aforementioned methods. It is often difficult to analyze the evolution of multi-physics fields and improve the computer efficiency.

A novel numerical method, i.e., spectral method [16], has recently been developed, which establishes the functional Hilbert spaces to overcome the infinite-dimensional characteristics of microwave heating model. By capturing the dominated dynamical characteristics, the PDE model can be transformed into a low-order one to improve the computational efficiency dramatically. However, it is difficult to directly derive the eigenfunctions, especially for the nonlinear differential operators with the non-homogeneous boundary conditions. In order to overcome the constraint of non-homogeneous Neumann boundary conditions, Zhong et al. [17] improve the spectral Galerkin method, which can obtain the analytical eigenvalues and eigenfunctions. Navarro et al. [18] apply the spectral method into the a cylindrical container subject to the radial microwave irradiation, but the thermal boundary condition is homogeneous. However, for the microwave heating model, the Neumann boundary condition is a particular case, which will facilitate to obtain analytic basis functions. In the practical engineering, the Debye media are usually exposed to the different surroundings, which means applying the mixed boundary conditions can accurately describe the thermal transport process. To the best of our knowledge, the analytical spectral method is not suitable for the spatial differential operator with the mixed boundary conditions because it is impossible to homogenize the boundary conditions. From the theory of functional space, there are always corresponding eigenfunctions regardless the boundary conditions are homogeneous or non-homogeneous. Fortunately, developments in data-driven techniques [19] provide a novel idea for generating the optimal subspace. Applying the ensemble of snapshots can derive the empirical eigenfunctions [20,21] in the fields of traditional heating engineering, but most of results do not consider the spatial differential operator with the mixed boundary conditions. Therefore, the proposed data-driven method will not only offer an efficient idea for model reduction of microwave heating process, but also provide a theoretical support for improving the spectral method.

The main contribution of this paper is to develop an approach of spatiotemporal model reduction, which is based on the data-driven method, for microwave heating process with the mixed boundary conditions. First of all, the general framework of data-driven based spatiotemporal model reduction is presented by combining the theory of time-space separation and relationship between thermodynamic and electromagnetic fields. Subsequently, the mechanism model of microwave heating process is presented by analyzing the heat transport equation, Maxwell's equations and Poynting's theorem. Then, the Karhunen–Loève Decomposition method is used for generating the optimal empirical eigenfunctions, which can transform the algebraic equations into the reduced-order ODE model. At last, the model reduction approach is applied to the process of microwave heating Debye medium, which has the mixed boundary conditions and temperature-dependent permittivity. The comparison studies demonstrate the efficacy of proposed methodology.

## 2. General Framework of Model Reduction

Our research effort is centered on eventually developing a spatiotemporal model reduction for the process of the microwave heating Debye medium. Due to the temperature-dependent permittivity, the numerical methods, such as, FEM and FDTD, will be used for analyzing the thermodynamic and electromagnetic fields, respectively. The computational complexity depends on the mesh grids, iterative methods and computing platform. It



is difficult to guarantee the model precision while maintaining a low level of computational complexity on the same platform. The idea in the work is to capture the dominant dynamic characteristics.

The microwave heating model can be divided into electromagnetic and thermodynamic sub-models. As we know, it is difficult to obtain the dominant dynamic characteristics of electromagnetic propagation due to the form of time-harmonic oscillations for the microwave propagation. Besides, the electromagnetic boundary condition also implies that it is impossible to transform the Maxwell's equations into a simpler expression. Fortunately, the traditional time/space separation method can effectively reduce the complexity of thermal sub-model because the spatiotemporal parameters can be approximated as the Fourier series. By using Galerkin method or approximated inertial manifold method, the spatial basis functions can be truncated or updated in order to facilitate the model reconstruction. However, the mixed boundary conditions make the spatiotemporal parameters difficult to separate. Therefore, a data-driven method will be used for obtaining the empirical eigenfunction and reconstructing the thermodynamic sub-model. By combining the relationship between the electromagnetic and thermodynamic fields on the microwave heating process, we present the general framework of data-driven based spatiotemporal model reduction, which can be shown in Figure 1.

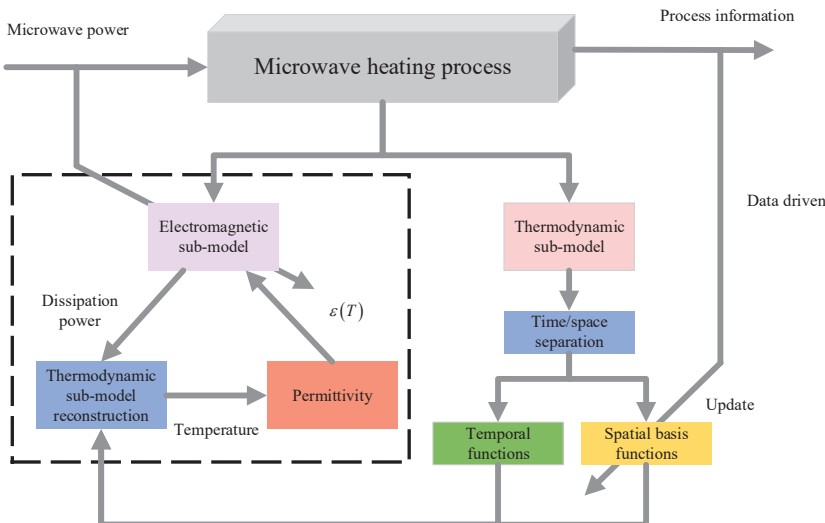

**Figure 1.** General framework of data-driven based spatiotemporal model reduction.

Based on the general framework in Figure 1, the main challenges are summarized as follows: Firstly, the thermodynamic process is usually described by parabolic PDEs with the mixed boundary conditions, whose infinite dimensional dynamics need to be approximated with finite dimensional ones to develop computationally efficient approaches. Secondly, the multi-physical fields vary significantly in the different regions, particularly in the regions of mixed boundary conditions, which increase the difficulty of time/space separation. Thirdly, it is difficult to directly reconstruct the thermodynamic sub-models because the eigenfunctions cannot be derived by applying the analytic method. Lastly, the serial computation between the Maxwell's equations and reduced-order ODE model needs to be further studied due to the temperature-dependent permittivity. To address the above problems, a data-driven based spatiotemporal model reduction approach is proposed to improve the computational efficiency for simulating the microwave heating process with the mixed boundary conditions.

## 3. Spatiotemporal Model Reduction for Microwave Heating Process

### 3.1. Mechanism Model of Microwave Heating Process

Microwave heating process is a complex process coupled with electromagnetic and thermodynamic mechanisms. As the aforementioned analysis, the electromagnetic sub-model

can be described by Maxwell's equations and Poynting's theorem. The complex dielectric constant usually depends on the local temperature and resonant frequency. It is worth pointing out that the microwave frequencies are fixed at 945 MHz and 2450 MHz. In the other words, the microwave propagation can be affected by not only the electromagnetic solution domain and boundary conditions, but also the temperature-dependent dielectric constant.

Different with the the traditional heating method, the microwave energy can be understood as the internal heat source, which will directly cause the rising of global temperature. The thermal conduction and thermal convection can facilitate the temperature uniformity. Besides, the temperature of medium differs significantly from those of the surroundings, which may lead to the complex boundary conditions and present a great challenge to numerical analysis. In order to simplify the complexity of multi-physics fields, the following five assumptions are considered:

**Assumption 1.** *The medium is linear, homogeneous, isotropic and nonmagnetic;*

**Assumption 2.** *The mass transfer is negligible;*

**Assumption 3.** *Initial temperature is the continuous and smooth profile;*

**Assumption 4.** *No volume changes are considered during heating;*

**Assumption 5.** *The dielectric constant is temperature dependent.*

Without loss of generality, the microwave heating model [22] can be described as follows:

$$\rho C_p \frac{\partial T}{\partial t} = \nabla \cdot (\kappa \nabla T) + Q(x, y, z, t) \tag{1}$$

subject to the mixed boundary conditions

$$\mathbf{n} \cdot \kappa \nabla T = h(T - T_\infty) + \sigma_h \varepsilon_h \left( T^4 - T_\infty^4 \right) \tag{2}$$

with the initial condition

$$T(x, y, z, 0) = T_0(x, y, z) \tag{3}$$

where $T$ denotes the global temperature in the different time $t$ and different positions $x, y, z$; $\rho$, $C_p$, $\kappa$, $T_\infty$, $h$, $\sigma_h$, $\varepsilon_h$ and $\mathbf{n}$ are the medium density, specific heat capacity, thermal conductivity, ambient temperature, heat transfer coefficient, Stefan Boltzmann constant, emissivity and the outward pointing unit normal on the surface of the medium. The dissipation power $Q(x, y, z, t)$, which is governed by the Poynting's theorem, can be expressed as

$$Q(x, y, z, t) = \frac{1}{2} \omega \varepsilon_0 \varepsilon'' \mathbf{E} \mathbf{E}^* \tag{4}$$

where $\omega$, $\varepsilon_0$ and $\varepsilon''$ are the angular frequency, vacuum permittivity and relative dielectric loss, respectively; $\mathbf{E}$ and $\mathbf{E}^*$ denote the electric field intensity and its complex conjugate, whose time-domain characteristics can be governed by the following Maxwell's equations:

$$\nabla \times \mathbf{H} = \frac{\partial \mathbf{D}}{\partial t} \tag{5}$$

$$\nabla \times \mathbf{E} = -\mu_0 \frac{\partial \mathbf{H}}{\partial t} \tag{6}$$

where $\mu_0$ denotes the vacuum permeability; $\mathbf{H}$ is the magnetic field; and $\mathbf{D}$ is the electric flux density, which can be expressed as

$$\mathbf{D}(\omega) = \varepsilon_0 \left( \varepsilon' - j\varepsilon'' \right) \mathbf{E}(\omega) \tag{7}$$

where $\varepsilon'$ is the relative dielectric constant.

*3.2. Data-Driven Based Spatiotemporal Model Reduction*

On the assumption that enough temperature sensors are uniformly placed into the Debye medium, we can obtain enough data with the spatiotemporal characteristics. Here, the data from the open-loop simulation can be defined as the snapshot $\hat{T}(x,y,z,t)$, which can be decomposed as

$$\hat{T}(x,y,z,t) = \sum_{X=1}^{\infty} \sum_{Y=1}^{\infty} \sum_{Z=1}^{\infty} \phi_X(x) \cdot \phi_Y(y) \cdot \phi_Z(z) \cdot \bar{T}(t) \tag{8}$$

where $\bar{T}(t)$ is only a function of $t$; $\phi_X(x)$, $\phi_Y(y)$ and $\phi_Z(z)$ is only the function assembles of $x$, $y$ and $z$, respectively. Let $\phi(x,y,z)$ ($\phi(x,y,z) = [\phi_X, \phi_Y, \phi_Z]$) be the matrix composed by the empirical eigenfunctions, which need to satisfy the following orthogonal properties

$$\phi_i^T(x,y,z)\phi_j(x,y,z) = \begin{cases} 0, & i \neq j \\ 1, & i = j \end{cases} \tag{9}$$

Obviously, it is impossible to directly obtain the empirical eigenfunctions. Motivated by Karhunen–Loève theorm, we can first construct a covariance matrix, which is defined as

$$\Sigma_T = \hat{T}(x,y,z,t) \cdot \hat{T}^T(x,y,z,t) - avg\hat{T}(x,y,z) \cdot avgT^T(x,y,z) \tag{10}$$

where $\hat{T}^T(x,y,z,t)$ denotes the transposition of vector; $avg\hat{T}(x,y,z)$ is the average of the snapshot $\hat{T}(x,y,z,t)$ at the different positions; it is noted that the covariance matrix $\Sigma_T = \Sigma_T^T$ is Hermitian. Using the singular value decomposition, the empirical eigenfunctions can be obtained as follows

$$\Sigma_T = U\Lambda V^T \tag{11}$$

where $U$ and $V$ are the left and right singular matrices; $\Lambda$ is the diagonal matrix $diag(\lambda_1, \lambda_2, \cdots, \lambda_n)$, which sorts the eigenvalue $\lambda_i$ from large to small. It is worth pointing out that the dominant dynamical characteristics can be captured by the low-dimensional eigenvalues, which can be expressed as

$$\Sigma_T \approx U_{n \times s} \Lambda_{s \times s} V_{s \times n}^T \tag{12}$$

where $|\lambda_{s+1}|/|\lambda_s| \to 0$. Based on the Galerkin truncation method, we can obtain the low-dimensional eigenfunctions, which can be defined as the ensemble of vectors $U_{n \times s} = [\phi_1(x,y,z), \phi_2(x,y,z), \cdots, \phi_s(x,y,z)] = \phi_S(x,y,z)$. With the assumption that the mechanism and data model can describe the same microwave heating process, the empirical eigenfunctions can be also used for decomposing (1)–(3). Therefore, (12) can be substituted into (1) such that

$$\phi_S(x,y,z) \cdot \rho C_p \dot{\bar{T}}_S(t) = \kappa \nabla^2(\phi_S(x,y,z)) \cdot \bar{T}_S(t) + Q(x,y,z,t) \tag{13}$$

where $\bar{T}_S(t)$ can be also called as the spectral function, which contains the dominated dynamical characteristics in spectral domain. Obviously, (13) can be simplified as

$$\dot{\bar{T}}(t) = \frac{\kappa}{\rho C_p} \phi_S^{-1}(x,y,z) \nabla^2(\phi_S(x,y,z)) \cdot \bar{T}_S(t) + \frac{1}{\rho C_p} \phi_S^{-1}(x,y,z) Q(x,y,z,t) \tag{14}$$

$$T(x,y,z,t) = \phi_S(x,y,z)\bar{T}_S(t) \tag{15}$$

**Remark 1.** *The eigenfunctions can be derived by the analytical method, whose precondition is that the boundary conditions can be homogeneous. However, it is difficult to transform the original model with the mixed boundary conditions into the one with homogeneous boundary condition. Different with the aforementioned results, the proposed data-driven method can directly obtain the*

*empirical eigenfunctions, which contain the dynamical characteristics of mixed boundary conditions. In the other words, the method can directly separate the spatiotemporal variables and obtain the low-dimensional ODE model, which will also overcome the restriction of mixed boundary conditions and decrease the computation burden.*

**Remark 2.** *Due to the sampling time and distribution of sensors, the snapshot is the discrete data ensemble, which may lead the discrete eigenfunctions. Therefore, the $\phi_S^{-1}(x, y, z)$ can be regarded as the generalized inverse of vector. Based on (9), we can obtain $\phi_S^{-1}(x) = \phi_S^T(x)$. Although the eigenfunctions can be obtained, it is difficult to obtain the second derivative of low-dimensional arrays from the view of numerical computation. If the eigenfunctions $\phi_S(x, y, z)$ can be fitted to obtain the analytical formulation, the spatial variables, i.e., x, y and z, may not be eliminated. Motivated by this consideration, the continuous differential operators need to be first transformed into the discrete one.*

Based on the explicit forward time and center space (FTCS) scheme [23], (1) can be written as the following set of algebraic equation:

$$\rho C_p \frac{T_{i,j,k}^{m+1} - T_{i,j,k}^m}{\Delta t} = \kappa \frac{T_{i+1,j,k}^m - 2T_{i,j,k}^m + T_{i-1,j,k}^m}{\Delta x^2} + \kappa \frac{T_{i,j+1,k}^m - 2T_{i,j,k}^m + T_{i,j-1,k}^m}{\Delta y^2}$$
$$+ \kappa \frac{T_{i,j,k+1}^m - 2T_{i,j,k}^m + T_{i,j,k-1}^m}{\Delta z^2} + Q_{i,j,k}^m \tag{16}$$

where superscript $m$ denotes the sampling time; subscript $i = 1, 2, \cdots, N$, $j = 1, 2, \cdots, N$ and $k = 1, 2, \cdots, N$ denote the nodes in the Cartesian coordinate system; $\Delta x$, $\Delta y$ and $\Delta z$ denote the sampling intervals. Moreover, the relationship between the sampling time $\Delta t$ and sampling interval $\Delta x$, $\Delta y$, $\Delta z$ needs to guarantee the stability of numerical computation and minimize the truncation error. Then, (16) can be simplified as the following explicit difference scheme:

$$\rho C_p T_{i,j,k}^{m+1} = \frac{\kappa \Delta t}{\rho C_p \Delta x^2} \left( T_{i+1,j,k}^m + T_{i-1,j,k}^m \right) + \left( \frac{1}{3} - \frac{2\kappa \Delta t}{\rho C_p \Delta x^2} \right) T_{i,j,k}^m$$
$$+ \frac{\kappa \Delta t}{\rho C_p \Delta y^2} \left( T_{i,j+1,k}^m + T_{i,j-1,k}^m \right) + \left( \frac{1}{3} - \frac{2\kappa \Delta t}{\rho C_p \Delta y^2} \right) T_{i,j,k}^m \tag{17}$$
$$+ \frac{\kappa \Delta t}{\rho C_p \Delta z^2} \left( T_{i,j,k+1}^m + T_{i,j,k-1}^m \right) + \left( \frac{1}{3} - \frac{2\kappa \Delta t}{\rho C_p \Delta z^2} \right) T_{i,j,k}^m + Q_{i,j,k}^m$$

In the process of microwave heating, the thermal radiation is so small that the external radiation term in (2) can be neglected. By applying the same difference method, the mixed boundary condition can be expressed as

$$\kappa \frac{T_{2,j,k}^m - T_{0,j,k}^m}{2\Delta x} - hT_{1,j,k}^m = -hT_\infty \tag{18}$$

$$\kappa \frac{T_{N+1,j,k}^m - T_{N-1,j,k}^m}{2\Delta x} + hT_{N,j,k}^m = hT_\infty \tag{19}$$

$$\kappa \frac{T_{i,2,k}^m - T_{i,0,k}^m}{2\Delta y} - hT_{i,1,k}^m = -hT_\infty \tag{20}$$

$$\kappa \frac{T_{i,N+1,k}^m - T_{i,N-1,k}^m}{2\Delta y} + hT_{i,N,k}^m = hT_\infty \tag{21}$$

$$\kappa \frac{T_{i,j,2}^m - T_{i,j,0}^m}{2\Delta z} - hT_{i,j,1}^m = -hT_\infty \tag{22}$$

$$\kappa \frac{T_{i,j,N+1}^m - T_{i,j,N-1}^m}{2\Delta z} + hT_{i,j,N}^m = hT_\infty \tag{23}$$

Based on (17)–(23), it is obvious that each node $T_{i,j,k}^m$ is only depended on the adjacent nodes from the view of numerical analysis. Substituting (18)–(23) into (17), some fictitious nodes, i.e., $T_{0,j,k}^m$, $T_{N+1,j,k}^m$, $T_{i,0,k}^m$, $T_{i,N+1,k}^m$, $T_{i,j,0}^m$ and $T_{i,j,N+1}^m$, can be eliminated and the discrete ODE can be derived. In the other words, the differential operator $\partial T/\partial t$ and $\nabla^2 T$ can be transformed as the weighted expression. Besides, the spatiotemporal variables $T_{i,j,k}^m$ can be also separated. On the definition of high-order vectors $T_D(x,y,z,m) = \left\{ T_{i,j,k}^m \right\}_{\forall i \forall j \forall k}$, the original PDE model can be rewritten as

$$T_D(x,y,z,m+1) = A T_D(x,y,z,m) + Q_D(x,y,z,m) + \omega_D(x,y,z,m) \tag{24}$$

where $A$ is a $N \times N \times N \times N$ dimensional sparse matrix, which is depended on the coefficients of spatiotemporal variables $\left\{ T_{i,j,k}^m \right\}_{\forall i \forall j \forall k}$ in (17)–(23); $Q_D(x,y,z,m)$ is the transient dissipation power; $\omega_D(x,y,z,m)$ can be regarded as the external disturbance, which is mainly depended on the ambient temperature. Based on the aforementioned empirical eigenfunctions $\phi_S(x,y,z)$, (24) can be transformed as

$$\begin{aligned}\bar{T}_{DS}(m+1) &= \phi_S^{-1}(x,y,z) A \phi_S(x,y,z) \bar{T}_{DS}(m) + \phi_S^{-1}(x,y,z) Q_D(x,y,z,m) \\ &+ \phi_S^{-1}(x,y,z) \omega_D(x,y,z,m)\end{aligned} \tag{25}$$

subject to the following initial condition:

$$\bar{T}_{DS}(0) \approx \phi_S^{-1}(x,y,z) T_D(x,y,z,0) \tag{26}$$

By defining $A_S = \phi_S^{-1}(x,y,z) A \phi_S(x,y,z)$, $Q_{DS}(m) = \phi_S^{-1}(x,y,z) Q_D(x,y,z,m)$ and $\omega_{DS}(m) = \phi_S^{-1}(x,y,z) \cdot \omega_D(x,y,z,m)$, (25) can be expressed as

$$\bar{T}_{DS}(m+1) = A_S \bar{T}_{DS}(m) + Q_{DS}(m) + \omega_{DS}(m) \tag{27}$$

**Remark 3.** *From the view of thermodynamics, the unstable temperature distribution is dependent on the different factors, such as, the external heat source, thermodynamic coefficients, boundary conditions and the forms of solution domain. The proposed methodology can obtain the spectral function of temperature distribution, which can facilitate to analyze the global characteristics. Therefore, the spectral functions of global dissipation power and mixed boundary condition need to be derived by applying the same empirical eigenfunctions $\phi_S(x,y,z)$. Different with the traditional Galerkin method, it is not necessary for the model reduction method to discuss the truncation error of parameters. The accuracy of model reduction is depended on the singular value decomposition (12) and finite difference method (17).*

**Remark 4.** *It is worth noting that the proposed method can successfully transform the thermodynamic sub-model into the ODE model based on the empirical eigenfunctions. However, the electromagnetic sub-model is still calculated by a FDTD method. From the view of mathematics, the data-driven method can also derive the basis functions of electromagnetic field, but it is difficult to obtain the local electromagnetic intensity inside the Debye media by using the sensors. Therefore, the traditional numerical method, such as, FDTD and FEM, are still applied to obtain the dissipation power. By combining the traditional method and data-driven method, the computational efficiency will be developed.*

It is inevitable for the finite difference method to generate the truncation error, so the accuracy of algebraic Equation (24) needs to be taken into account. As the proposed methodology discretizes spatiotemporal parameters and reduces the order of empirical eigenfunctions, there will be model mismatches. Therefore, it is necessary to given a

trade-off between the accuracy of spatiotemporal synthesis and computational efficiency, whose residual evaluation can be used:

$$\sigma = \frac{(T(x,y,z,m) - T_D(x,y,z,m))^2}{T^2(x,y,z,m)} \times 100\% \tag{28}$$

Based on aforementioned analysis, the implementation of the proposed algorithm (Algorithm 1) is explicitly explained shown below:

---

**Algorithm 1** Detailed algorithmic procedure of model reduction for microwave heating Debye media.

---

**Initialize:** Parameters of thermodynamics and electromagnetism $\rho$, $C_p$, $\kappa$, $h$, $\varepsilon_0$, $\varepsilon'(T)$, $\varepsilon''(T)$, incident electric field $E_0$ and sampling time $\Delta t$;

1: Mesh solution domain of electromagnetic and thermodynamic sub-model based on the sampling interval $\Delta x$, $\Delta y$, $\Delta z$ and discrete nodes $N$;
2: Construct the sparse matrix $A$;
3: Obtain the sufficient snapshot $\hat{T}(x,y,z,m)$ and initial condition $T_0(x,y,z)$;
4: Derive the appropriate empirical eigenfunctions $\phi_S(x,y,z)$;
5: **For** $m = 1 : M$ **do**
6:   Solve the time-dependent Maxwell's Equations (5) and (6) to obtain the local electric field intensity and phase;
7:   Calculate the local transient dissipation power $Q(x_i, y_j, z_k, m)$ based on (4);
8:   Apply the linear approximation and compute the global transient dissipated power $Q(x,y,z,m)$;
9:   Update $\omega_D(x,y,z,m)$ based on the ambient temperature $T_\infty$;
10:   Substitute $Q_D(x,y,z,m)$ and $\omega_D(x,y,z,m)$ into (27) and obtain temperature spectra $\hat{T}_{DS}(m)$;
11:   Synthesize the spatiotemporal parameter $T_D(x,y,z,m)$ and update the dielectric constant;
12:   Compare the residual evaluation $\sigma$ with the expect residual evaluation $\sigma_{exp}$;
13:   **If** $\sigma < \sigma_{exp}$ **then**
14:     Go to Step 5;
15:   **else**
16:     $s = s + 1$, go to Step 4;
17:   **End**
18: **End**

---

## 4. Simulation and Analysis

In this section, the proposed model reduction method is applied to decompose the PDE model of microwave heating Debye media. Based on the aforementioned analysis, we can obtain that the heat transport in the orthogonal 3D Cartesian coordinate system can be regarded as the independent spatiotemporal coupled process. In other words, the thermodynamic sub-model is linear, which means that the 3D temperature distribution is generated by superimposing on the thermal conduction in the different coordinate axis. Besides, the proposed model reduction method is also based the orthogonal empirical eigenfunctions (9), which indicate that 1D and 2D conditions can also demonstrate the effectiveness of proposed methodology. In order to facilitate the demonstration, we only consider the coupled process of electromagnetic and thermodynamic fields in the x-axis, whose schematic is shown in Figure 2.

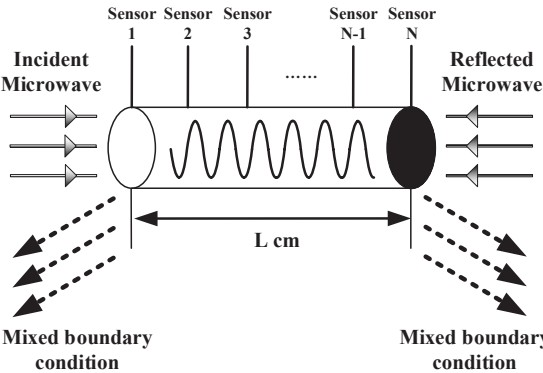

**Figure 2.** Detailed schematic for microwave heating model.

As Figure 2 shown, the Debye medium is exposed by the uniform transverse electromagnetic (TEM) wave, whose incident angle is in a perpendicular to the left surface. The right surface is the perfect boundary condition, in which the reflection coefficient can be denoted as 1. Based on the Maxwell's Equations (5) and (6), the microwave propagation sub-model can be simplified as

$$\frac{d^2 E_x}{dz^2} + \left(\frac{2\pi f}{c}\right)^2 \varepsilon(T)E_x = 0 \tag{29}$$

where $f$ denotes the resonant frequency of microwave 2.45 GHz and $c$ denotes the speed of light $3.0 \times 10^8$ m/s. By using the FDTD method, the solution of electric field $E_x$ in (29) can be obtained. With the Poynting's theorem (4), we can obtain the global dissipation power, which can be used for the non-homogeneous term of heat transport equation. It is worth pointing out that the formulation of dissipation power can be expressed as the $N \times 1$ vector. Different with the aforementioned results [17], the dissipation power does not be transformed as the explicit mathematical formulation, which not only improve the computing speed, but also decrease the error of nonlinear fitting. Without loss of generality, the de-ionized water is chosen as the typical medium due to the relative wider range of permittivity spectrum. According to [24], the relative complex dielectric constant $\varepsilon(T)$ is depended on the following equation:

$$\varepsilon(T) = \varepsilon'(T) - j\varepsilon''(T) = 5.5 + \frac{82.6 - 0.403T + 0.0009T^2}{1 + 2.45j \big/ 0.168(T + 22.05)^{1.23}} \tag{30}$$

Obviously, the dissipation power can be derived if the temperature in each node can be obtained. Subsequently, we will investigate the 1D thermodynamic sub-model, which can be simplified as

$$\frac{\partial T(x,t)}{\partial t} = \frac{\kappa}{\rho C_p} \frac{\partial T^2(x,t)}{\partial x^2} + \frac{1}{\rho C_p} Q(x,t) \tag{31}$$

subject to the mixed boundary conditions

$$\kappa \frac{\partial T(1,t)}{\partial x} - hT(1,t) = -hT_\infty$$
$$\kappa \frac{\partial T(l,t)}{\partial x} + hT(l,t) = hT_\infty \tag{32}$$

where $C_p = 4.2$ J/(g·°C), $\kappa = 0.0054$ W/(cm·°C), $h = 0.005$ W/(cm$^2$·°C) and $\rho = 1$ g/cm$^3$. On assumption that the depth of medium $L$ is 6 cm, initial temperature $T(x,0)$ is 0 °C, ambient temperature $T_\infty$ is 20 °C and the intensity of incident electric field $E_0$ is 1 V/cm, the temperature profiles can be obtained by applying the uniformly-spaced sampling interval 0.01 cm and sampling time 0.01 s. It is a remarkable fact that

the media is subject to the mixed boundary conditions, which mean that the empirical eigenfunctions need to be derived by applying the Algorithm 1. Based on the aforementioned method, the FTCS scheme will be used for discretizing the 1D governed Equation (31), which can be simplified as

$$T(x_i, t_{m+1}) = \gamma_1[T(x_{i+1}, t_m) + T(x_{i-1}, t_m)] + (1 - 2\gamma_1)T(x_i, t_m) \tag{33}$$

and the mixed boundary condition can also be transformed as

$$T(x_0, t_m) = T(x_2, t_m) - \frac{2h\Delta x}{\kappa}T(x_1, t_m) + \frac{2h\Delta x}{\kappa}T_\infty \tag{34}$$

$$T(x_{N+1}, t_m) = T(x_{N-1}, t_m) - \frac{2h\Delta x}{\kappa}T(x_N, t_m) + \frac{2h\Delta x}{\kappa}T_\infty \tag{35}$$

Substituting (34) and (35) into (33), the recursive equations of boundary node can be expressed as

$$T(x_1, t_{m+1}) = 2\gamma_1 T(x_2, t_m) + 2\gamma_2 T_\infty + (1 - 2\gamma_1 - 2\gamma_2)T(x_1, t_m) \tag{36}$$

$$T(x_N, t_{m+1}) = 2\gamma_1 T(x_{N-1}, t_m) + 2\gamma_2 T_\infty + (1 - 2\gamma_1 - 2\gamma_2)T(x_N, t_m) \tag{37}$$

where $\gamma_1 = \kappa\Delta t/(\rho C_p\Delta x^2)$ and $\gamma_2 = h\Delta t/\rho C_p\Delta x$. By combining (33), (36) and (37), the 1d thermodynamic sub-model can be transformed as the following discrete formulation

$$
\begin{bmatrix} T(x_1, t_{j+1}) \\ T(x_2, t_{j+1}) \\ \vdots \\ Q(x_{N-1}, t_j) \\ T(x_N, t_{j+1}) \end{bmatrix} = \begin{bmatrix} 2\gamma_2 T_\infty + Q(x_1, t_j) \\ Q(x_2, t_j) \\ \vdots \\ Q(x_{N-1}, t_j) \\ 2\gamma_2 T_\infty + Q(x_N, t_j) \end{bmatrix}
$$
$$
+ \begin{bmatrix} \gamma' & 2\gamma_1 & & & \\ \gamma_1 & 1-2\gamma_1 & \gamma_1 & & \\ & \cdots & \cdots & \cdots & \\ & & \gamma_1 & 1-2\gamma_1 & \gamma_1 \\ & & & 2\gamma_1 & \gamma' \end{bmatrix} \begin{bmatrix} T(x_1, t_j) \\ T(x_2, t_j) \\ \vdots \\ T(x_{N-1}, t_j) \\ T(x_N, t_j) \end{bmatrix} \tag{38}
$$

where $\gamma' = 1 - 2\gamma_1 - 2\gamma_2$. It is obvious that (38) has transform the temporal differential operator, spatial differential operator and mixed boundary conditions into the matrix formulations. As Figure 2 shown, $N$ sensors are uniformly placed to measure the $N$ temperature rise curves. By implementing the open-loop simulation of (4), (29), (30), (38), the snapshots of microwave heating Debye media can be obtained. On the assumption the sampling times is chosen as $M$, the snapshots can be expressed as the ensemble of $\hat{T}(x, t)$, which can be defined as

$$
\hat{T}(x, t) = \begin{bmatrix} \hat{T}(x_1, t_1) & \hat{T}(x_1, t_2) & \cdots & \hat{T}(x_1, t_M) \\ \hat{T}(x_2, t_1) & \hat{T}(x_2, t_2) & \cdots & \hat{T}(x_2, t_M) \\ \vdots & \vdots & \cdots & \vdots \\ \hat{T}(x_N, t_1) & \hat{T}(x_N, t_2) & \cdots & \hat{T}(x_N, t_M) \end{bmatrix} \tag{39}
$$

With the singular value decomposition (11), the empirical eigenvalues and eigenfunctions can be obtained. The empirical eigenvalues are sorted in a descending order and the normalized cumulative sum of eigenvalues ($\sum_{i=1}^{r}\lambda_i \big/ \sum_{i=1}^{m}\lambda_i$) can be obtained. It is obvious that 99.99% of the dynamic characteristics are embedded in the first six eigenfunctions. In order to capture the more dominated dynamic characteristics, we choose the first six eigenfunctions $\phi_S(x)$, which is described as the six vectors instead of the analytical function. Besides, (12) is

also hold with the low-dimensional eigenvalues and eigenfunctions. In fact, the empirical eigenfunctions have the spatial characteristics, which can be shown in Figure 3.

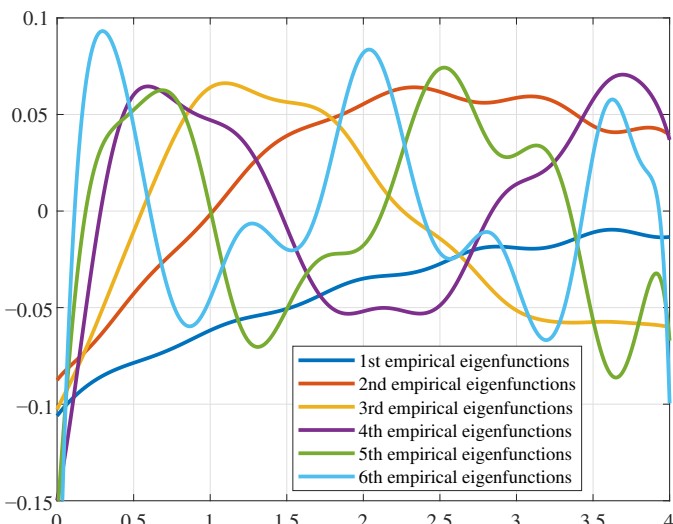

**Figure 3.** The first six empirical eigenfunctions based on the microwave heating Debye media.

Substituting $\phi_S(x)$ into (38), the reduced-order model can be obtained. Based on the temperature-dependent permittivity (30), the electromagnetic and thermodynamic sub-model can be updated and the spatiotemporal parameters $T_D(x,m)$ can be obtained. On the assumption that the sampling times $M$ are 10,000, we can obtain the global temperature distribution, which is shown in Figure 4.

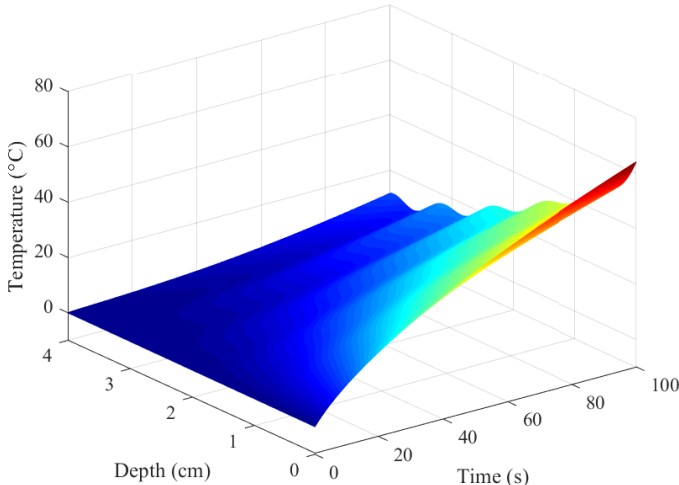

**Figure 4.** Global temperature distribution for the reduced-order model with microwave heating Debye media.

In order to demonstrate the effectiveness of proposed methodology, we compare the numerical results of reduced-order model with the original model. It is worth pointing that the same thermodynamics and electromagnetic conditions are introduced in the original model whose numerical solution will be derived by applying the traditional FDTD method. On the assumption that the results of original model can be regarded as the benchmark, we can obtain the error of global temperature distribution, which can be shown in Figure 5. Furthermore, we also choose the different times (i.e., 100 s, 80 s, 60 s, 40 s and 20 s) and different locations (i.e., 4 cm, 3 cm, 2 cm, 1 cm and 0 cm) to compare the temperature variations, whose results are shown in Figure 6 and Figure 7, respectively. It is obvious that the agreement of global

temperature distribution is satisfactory because the absolute error is not more than 0.03, which can satisfy the requirement of expect residual evaluation $\sigma_{\text{exp}}$. In terms of computation time, the average elapsed time of reduced-order model only costs $1.438 \times 10^{-5}$ s using a laptop computer with i5-6200U CPU and 8G RAM. Compared with the elapse time of original model ($2.2868 \times 10^{-4}$ s) in each sampling time, the computation efficiency of reduced-order model has been improved by approximately 16 times using the same computer. Therefore, the proposed methodology can significantly develop the computation speed in the premise of guaranteeing the predicted accuracy of spatiotemporal parameters.

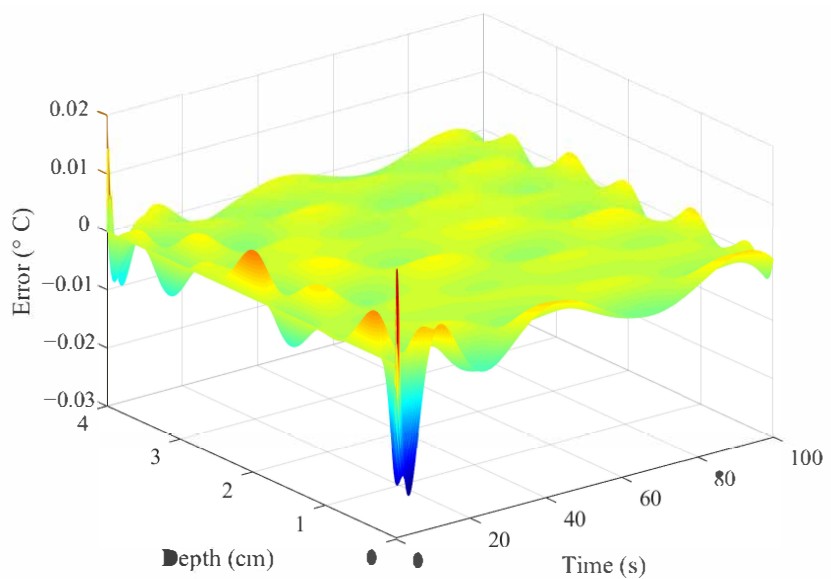

**Figure 5.** Comparison between the reduced-order model and original model for the global temperature distribution.

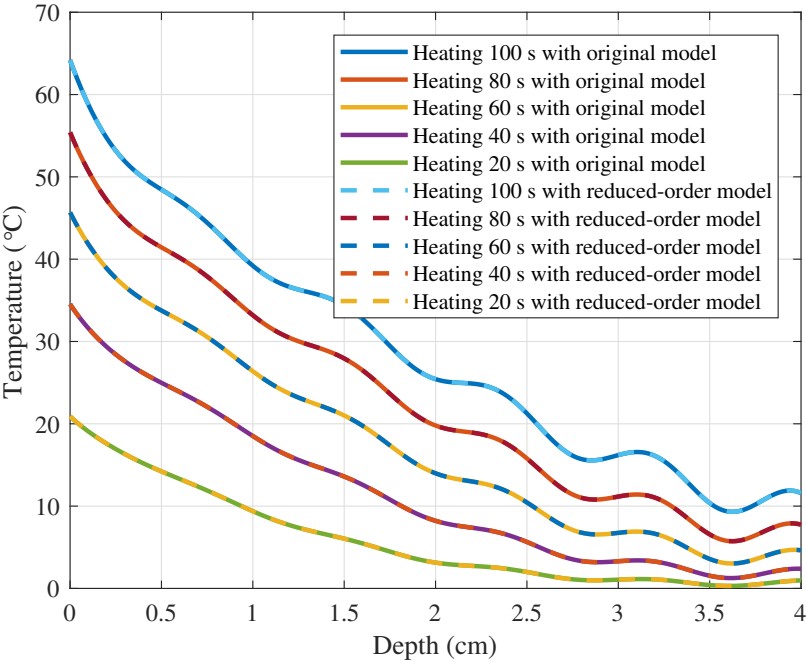

**Figure 6.** Comparison between the reduced-order model and original model for the spatial temperature distribution at 100 s, 80 s, 60 s, 40 s and 20 s.

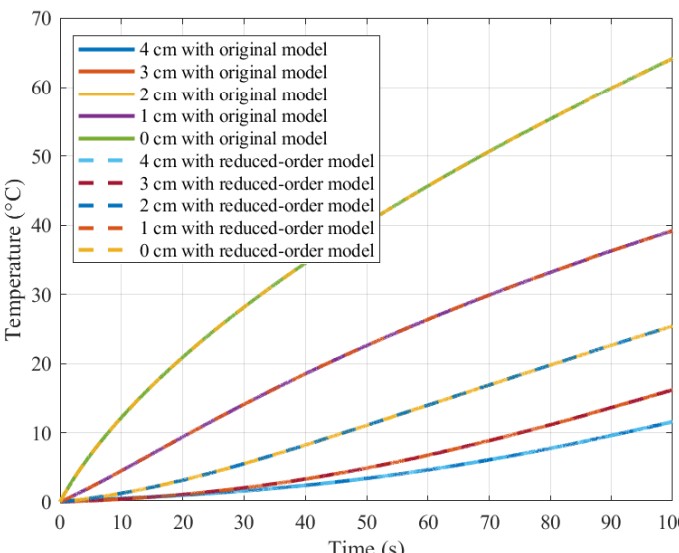

**Figure 7.** Comparison between the reduced-order model and original model for the temperature rise curves at 4 cm, 3 cm, 2 cm, 1 cm and 0 cm.

## 5. Conclusions

We present a model reduction approach for microwave heating process described by the PDE model subject to the mixed boundary conditions. The basic idea of this paper is to obtain the empirical eigenfunctions which are generated by applying Karhunen–Loève decomposition. It is important to obtain the proper snapshots based on the open-loop simulation or experiments. In addition, the traditional PDE model needs to be discretized as the recursive algebraic equations. With the help of matrix manipulation and Galerkin truncation method, we can obtain the reduced-order model which can capture the dominant dynamical characteristics. The proposed approach is applied to the process of microwave heating Debye media. The comparison results between the reduced-order model and original model demonstrate the effectiveness of the proposed methodology. Further studies are underway for designing the experiments to obtain the snapshots of temperature distribution, which can derive a reduced-order model to describe the actual process of microwave heating Debye media.

**Author Contributions:** Conceptualization, J.Z. and S.L.; methodology, J.Z. and S.L.; software, J.Z.; validation, J.Z.; formal analysis, J.Z.; investigation, J.Z. and S.L.; resources, J.Z.; data curation, J.Z.; writing—original draft preparation, J.Z.; writing—review and editing, J.Z. and S.L.; visualization, J.Z.; supervision, S.L.; project administration, J.Z.; funding acquisition, J.Z. and S.L. All authors have read and agreed to the published version of the manuscript.

**Funding:** This research was funded by National Natural Science Foundation of China (62003066, 61771077), the Science and Technology Research Program of Chongqing Municipal Education Commission (KJQN201900614) and the China Scholarship Council (202008500013).

**Institutional Review Board Statement:** Not applicable.

**Informed Consent Statement:** Not applicable.

**Data Availability Statement:** The datasets analysed during the current study are available from the corresponding author on reasonable request.

**Acknowledgments:** The authors would like to thank J. Bao for the critical comments.

**Conflicts of Interest:** The authors declare no conflict of interest.

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
