# Peer review of "A Data-Driven Based Spatiotemporal Model Reduction for Microwave Heating Process with the Mixed Boundary Conditions"

_processes, doi:10.3390/pr9050827_

Round 1

Reviewer 1 Report

The authors are claiming to present a novel data-driven spatial temporal model reduction that is particularly suitable for microwave heating. The authors creates the impression to simplify the calculation of the multi-physics problem by introduction of a set of orthogonal eigenfunctions. To the understanding of the reviewer, this is not the case. The reviewers are expanding the profile of the temperature over a limited volume with certain boundary conditions. The electromagnetic field is still calculated by a FDTD method. Hence, the authors must explain why the method is specifically focusing on microwave heating. The authors are must explain the validity of the expansion method. The authors are separating the temperature profile in 3 orthogonal functions for the 3-dimensional problem first. This is possible. But, to the understanding of the reviewer, an expansion into orthogonal eigenfunctions requires a lossless structure, hence the structure must be closed by lossless walls to calculate a proper set of eigenfunctions. The material inside must be lossless too. In a second step currents must be introduced that describe the boundary conditions and the losses. The authors try to validate their method by a certain cylindrical structure and showing the first 6th empirical found eigenfunctions. It is not compared to simulation done by e.g. Comsol and experiment. This is not sufficient. The authors must be precise in the abstract and the introduction what is done and why the method is particularly suitable for microwave heating.

Author Response

Dear Reviewer,

Thanks very much for your professional comments regard to our manuscript (processes-1177383). According to the comments, we have tried our best to check the paper and address all the comments. The changes are highlighted in PDF file. And the notes have been attached. Please see the attachment.

We appreciate for Editors/Reviewers’ professional works for the manuscript, and hope that the revised manuscript could satisfy the requirements for publication in this journal. Once again, thank you very much for your valuable comments and suggestions.

If you have any question about this paper, please do not hesitate to let me know.

Kind Regards,

Sincerely yours,

Jiaqi Zhong

Reviewer 2 Report

It is an interesting work with a new approach about modelling the coupled electromagnetic-thermal problem for microwave heating purposes. However its application to 3D real models should be discussed in depth. Experimental measurements to verify the model are needed at some point, now or in the future.

Author Response

(The authors gave the same response as above.)

Round 2

Reviewer 1 Report

Many thanks, the manuscript required the update. Still the title and content is confusing, as it focuses on the thermal aspects. The microwave heating is irrelevant here. The introduction of thermal heat could be applied in general.  To the opinion of the reviewer it reduces the relevance of the manuscript significantly. 

Author Response

Point 1: Many thanks, the manuscript required the update. Still the title and content is confusing, as it focuses on the thermal aspects. The microwave heating is irrelevant here. The introduction of thermal heat could be applied in general. To the opinion of the reviewer it reduces the relevance of the manuscript significantly.

Response 1: Thank you very much for your professional comments. We have removed some irrelevant introduction of traditional heating process, such as,

“The linear/nonlinear transformation methods can be applied to map the time-domain parameters into the frequent-domain ones [17,18], which can explicitly reveal the principle of multi-physics coupling.”

“In other words, the eigenfunctions are always embedded in the ensemble of snapshots [21,22]. The data-driven method can overcome the deficiency of traditional spectral method and transform the PDE model with the mixed boundary conditions into the reduced-order model.”

And we also provide the meaning of the proposed spectral method in the latest version of the manuscript, as shown:

“Applying the ensemble of snapshots can derive the empirical eigenfunctions [20, 21] in the fields of traditional heating engineering, but most of results do not consider the spatial differential operator with the mixed boundary conditions. Therefore, the proposed data-driven method will not only offer an efficient idea for model reduction of microwave heating process, but also provide a theoretical support for improving the spectral method.”

Reviewer 2 Report

Authors have addressed my comments correctly

English should be reviewed

Author Response

Point 1: Authors have addressed my comments correctly. English should be reviewed

Response 1: Thanks once again for your useful comments. We have tried our best to check the whole manuscript and the readability has been improved.